# The Correlation between the Vascular Calcification Score of the Coronary Artery and the Abdominal Aorta in Patients with Psoriasis

**DOI:** 10.3390/diagnostics13020274

**Published:** 2023-01-11

**Authors:** Trang Nguyen-Mai Huynh, Fumikazu Yamazaki, Izumi Kishimoto, Akihiro Tanaka, Yonsu Son, Yoshio Ozaki, Kazuya Takehana, Hideaki Tanizaki

**Affiliations:** 1Department of Dermatology, Kansai Medical University, Osaka 573-1010, Japan; 2Psoriasis Center, Kansai Medical University, Osaka 573-1191, Japan; 3Division of Rheumatology, Department of Medicine I, Kansai Medical University, Osaka 573-1010, Japan; 4Division of Cardiology, Department of Medicine II, Kansai Medical University, Osaka 573-1010, Japan

**Keywords:** calcification, coronary artery, abdominal aorta, psoriasis, risk factors, velocity, cardiovascular diseases

## Abstract

Psoriasis is known as an independent risk factor for cardiovascular disease due to its chronic inflammation. Studies have been conducted to evaluate the progress of atherosclerotic plaques in psoriasis. However, inadequate efforts have been made to clarify the relationship between atherosclerosis progress in coronary arteries and other important blood vessels. For that reason, we investigated the correlation and development of the coronary artery calcification score (CACS) and the abdominal aortic calcification score (AACS) during a follow-up examination. Eighty-three patients with psoriasis underwent coronary computed tomography angiography (CCTA) for total CACS and abdominal computed tomography (AbCT) for total AACS. PASI score, other clinical features, and blood samples were collected at the same time. The patients’ medical histories were also retrieved for further analysis. Linear regression was used to analyze the CACS and AACS associations. There was a moderate correlation between CACS and AACS, while both calcification scores relatively reflected the coronary plaque number, coronary stenosis number, and stenosis severity observed with CCTA. Both calcification scores were independent of the PASI score. However, a significantly higher CACS was found in psoriatic arthritis, whereas no similar phenomenon was recorded for AACS. To conclude, both CACS and AACS might be potential alternative tests to predict the presence of coronary lesions as confirmed by CCTA.

## 1. Introduction

Currently, 1% of people worldwide suffer from psoriasis, one of the most common skin diseases. In addition to skin inflammation, cardiovascular disease has been identified as the leading cause of death in patients with psoriasis This latent inflammation may not only harm the appearance of those who are suffering, but it may also be a contributing factor to atherosclerosis disease. The evaluation of the progression of cardiovascular burden now includes not only atherosclerosis but also vascular calcification. Several studies have clarified that vascular calcification might be derived from both systemic inflammatory reactions and mineral reaction abnormalities occurring in patients with psoriasis [1]. Different instruments have been compared to identify the most helpful method to evaluate calcification lesions, such as coronary CT angiography (CCTA), computed tomography (CT), coronary MRI, invasive coronary angiography (ICA), and intravascular ultrasound (IVUS) [2,3]. CCTA could be chosen for calcification lesions as a first-line invasive modality [4]. The blood coronary arteries and abdominal aorta in particular have been measured in a few studies to shed light on the most vulnerable areas that might lead to sudden all-cause death as well as cause-specific death [5,6,7].

Using CCTA, we previously reported that cardiovascular abnormalities were more common in 88 Japanese patients with psoriasis before the use of biologic agents than in healthy controls [8]. However, CCTA is not a simple test, even though it is effective in detecting cardiovascular involvement in psoriasis. Based on the limitations of CCTA, a simpler method is suggested for predicting the presence of coronary artery lesions. To accomplish this goal, we selected the coronary artery calcification score (CACS), which is determined by using a simpler 3 mm chest CT, to verify its correlation with CCTA.

The abdominal aortic calcification score (AACS) has been found to correlate with cardiovascular pathology in patients with diabetes mellitus or renal failure [9,10,11], although it does not share a correlation with cardiovascular pathology in healthy subjects. Since patients with psoriasis have long-term systemic inflammation, we suspected the existence of a correlation between coronary artery calcification status and abdominal aortic calcification progression as well as, a correlation between cardiovascular pathology and AACS. Therefore, we further examined the AACS of patients with psoriasis who underwent CCTA in a previous study and performed a paired comparison of each psoriatic patient’s coronary and abdominal artery calcification scores based on CT scanner results.

## 2. Materials and Methods

### 2.1. Study Design

The data of 203 consecutive visits of 89 patients with psoriasis to the Psoriasis Center, Kansai Medical University Affiliated Hospital were obtained for the period of 2015 to 2019. Six patients who could not give consent were excluded from the study. Of the 83 patients with psoriasis who participated at the time of examination, 73 patients completed at least one CCTA and one AbCT, of which 64 patients completed both within 24 months, and the remaining patients underwent these two scans at intervals longer than 24 months (ranging from 1 month to 67 months). Of these, 15 patients completed at least 2 CCTAs, and 20 patients completed at least 2 AbCTs (Figure 1).

Data retrieved from patients included the following details: date of the first visit, age of onset, and PASI scores before starting specific treatment. Patients with institutional review board approval and written informed consent were undergoing routine clinical and laboratory examinations. Patients were re-examined after about 1–3 years to evaluate the progress of their CACS and AACS. At that time, relevant data were also collected (Figure 2). The study protocols complied with the Declaration of Helsinki and were approved by the Institutional Review Board of Kansai Medical University, Osaka, Japan.

### 2.2. Demographics, Clinical Measurements, and Laboratory Measurements

Demographic data and covariates were all accessed at the time of examination. Demographic information, including age, gender, lifestyle habits, comorbidities, and records of family coronary and psoriasis history, was self-reported via a questionnaire. Patients were classified as smokers or alcohol consumers if they were current consumers or used to consume them, while patients were classified as nonsmokers or nonalcoholics if they had never consumed them. A physical examination was performed by a clinical physician to collect data on BMI, blood pressure, psoriasis area severity index (PASI score), psoriasis duration, Suita score, Framingham risk score, coronary heart disease (CHD) after 10 years of risk score, and current specific treatment of psoriasis. Body mass index (BMI; kg/m^2^) was calculated as body weight (kg) divided by squared height (m^2^). A sitting rest blood pressure of 140/90 mmHg or higher, or current treatment with antihypertensive drugs, was considered to indicate hypertension. Diabetes mellitus was defined as a fasting plasma glucose test of 126 mg/dL or more, a 2 h glucose tolerance test of 200 mg/dL or more, hemoglobin A1c of 6.5% or more, or the current use of antidiabetic medication. Dyslipidemia was defined as a plasma triglyceride of 150 mg/dL or more, a plasma high-density lipoprotein cholesterol of 40 mg/dL or less, a plasma low-density lipoprotein cholesterol of 140 mg/dL or more, or current treatment for dyslipidemia (according to the 2017 guideline published by the Japan Atherosclerosis Society). Participants’ BMIs were classified into five subcategories based on the World Health Organization (WHO) classification for Asians as follows: <18.5, underweight; 18.5–<23.0, normal; 23–<25.0, overweight; 25.0–<30.0, obesity class I; and ≥30.0, obesity class II. Metabolic syndrome patients were diagnosed as negative, undefined, or positive if patients with psoriasis met 0, 1–2, and ≥3 criteria, respectively, according to the National Cholesterol Education Program (NCEP) Adult Treatment Panel III (ATP III) and WHO for Asians, as shown in the Table 1.

Hypertension, diabetes mellitus, and hyperlipidemia statuses of the participant were separately collected as independent covariates during data analysis due to their complicated and critical roles in artery pathophysiology. The cardiovascular disease (CVD) complications listed in this study included coronary heart disease (CHD), stroke, arrhythmia, and aortic aneurysm. Participants in this study had CHD complications, such as stable angina, prior myocardial infarction, and coronary revascularization. Various diseases in patients with psoriasis that were not related to the aim of this study, such as acute nephropathy, cirrhosis, colon polyps, stomach cancer, malignant lymphoma, renal failure, and Takotsubo cardiomyopathy, were also categorized as belonging to the subgroup “Other” in the comorbidity variable.

Oral psoriasis treatment was defined as the use of the following drugs: methotrexate, cyclosporine, and apremilast. Anti-TNF alpha drugs were prescribed in this study, including adalimumab and infliximab. Anti-IL17A drugs, including ixekizumab, secukinumab, and brodalumab and anti-IL12/23 drugs, including ustekinumab, were also the current treatment options for some patients in the study.

Additional data from blood samples, including plasma total cholesterol, plasma triglycerides, plasma high-density lipoprotein cholesterol, plasma low-density lipoprotein cholesterol, fasting plasma glucose, plasma hemoglobin A1c, and plasma high-sensitive C-reactive protein levels, were collected following appropriate standardized protocols.

Missing covariate values were estimated by carrying backward the most recent exam value for smoking and alcohol status and carrying forward recent values for psoriasis and coronary family history.

### 2.3. Measurement of CACS and AACS via CT Scan

A total of 83 patients underwent CCTA on the same day as the blood withdrawal. The same 128-row multidetector CT scanner machine, SOMATOM Definition Flash, was utilized for both CCTA and AbCT, with different protocols being followed for each procedure as shown in the Table 2.

The Agatston score was used to quantify calcification status in both coronary arteries and the abdominal aorta because of its evidence-based accuracy [12,13]. Plaques of interest at a given artery were drawn by trained physicians who were blinded to the clinical outcomes of participants. The Agatston score of each lesion was then automatically calculated using ZIO Station 2 Plus software, which multiplied the volume (mm^3^) by the density score of the plaque (according to the Hounsfield unit of plaque). The Agatston score was corrected for slice thickness and calculated for a minimum volume of 1.06 mm^3^ (1 boxel) with an attenuation threshold of 130 Hounsfield units or more. Due to evidence from previous studies [14,15], the 130-Hounsfield threshold attenuation value was chosen for artery calcification.

Three major branches of the coronary artery, including the left anterior descending (LAD), left circumflex (LCx), and right coronary artery (RCA), were the targets of the Agatston score assessment. Moreover, all coronary segments were also objectively analyzed by trained physicians in terms of total plaque number, total stenosis number, stenosis severity, stenosis distribution, and noncalcified plaque burden.

The coronary stenosis severity and distribution, which were optimized from a previous publication [16], were visually assessed as discussed in the Table 3.

For the abdominal CT scan image, supine image data were used. The Agatston score assessment for the abdominal aorta was performed under similar procedures from 1 cm above the origin of the celiac artery to 1 cm above the iliac bifurcation. Then, 15–20 slices of abdominal–pelvic CT scan images were taken for each participant, with a slice thickness of 5 mm.

The total Agatston score across coronary arteries and the whole section of the abdominal aorta were then summed to obtain CACS and AACS, respectively, for each participant.

### 2.4. Statistical Analysis

The mean, standard deviation, and median with an interquartile range were used to report continuous variables. To assess the normality distribution, skewness, and kurtosis measurements, Kolmogorov–Smirnov and Shapiro–Wilk tests (for N ≥ 50 and N < 50, respectively) were used. The Student’s *t*-test for parametric data and one-way ANOVA test were performed for parametric continuous variables within two subgroups. For nonparametric continuous variables with more than two subgroups, the Mann–Whitney U test and the Kruskal–Wallis test were used. A two-way ANOVA was also performed to examine the effects of two continuous independent variables.

Categorical variables were summarized as percentages, and Pearson’s χ^2^ test was performed for these variables. Furthermore, for longitudinal follow-up data, continuous variables were assessed with a paired *t*-test or a Wilcoxon matched-pairs signed-rank test.

Associations between calcium scores and other continuous variables were measured using the Spearman rank correlation coefficient. Multivariable linear regression analyses were then conducted to evaluate the relationship between the change in calcification scores and the change in other relative continuous variables. Simple logistic regression analysis was used to evaluate the value of CACS and AACS for predicting critical characteristics, including coronary plaque, coronary stenosis, noncalcified plaque, and comorbidity, which might contribute to the all-cause mortality of patients with psoriasis. ROC curves were then plotted with relative sensitivity and specificity percentages. The AUC was calculated to compare the prediction values of CACS and AACS.

To avoid interscan biases, we performed the Hokanson method in order to define a significant change in CAC as the following equation $\left|\sqrt{CAC2}-\sqrt{CAC1}\right|>2.5$ [17,18]. Generally, the progression of CACS and AACS has been classified into 2 categories: (1) CAC/AAC incidence for those who initially had CAC/AAC = 0 and then > 0 at the end of the study, and (2) CAC/AACS progression for those who initially had CAC/AAC > 0 [19]. Among 10 individuals who initially had CAC = 0, 2 patients had CAC > 0. After recalculating with the Hokanson method, 1 patient was considered to have a significant change in CAC during follow-up. Hence, the CAC incidence rate was 10%. Among the 10 individuals who had CAC > 0 at the beginning of our study, after readjusting with the Hokanson method, it was observed that 4 patients had significant changes in CACS. Thus, the CACS progression rate was 40%. We calculated both the absolute change and the relative change in CAC and divided the time between CT scans by months. The absolute CAC progression was measured as follows: (CAC2−CAC1)/month_interval. The relative CAC progression was measured as follows: (ln(CAC2+25)−ln(CAC1+25))/month_interval [19]. Similar methods were used for AAC. Among the 5 individuals who had AAC = 0 at the first AbCT, 4 remained at AAC = 0 at the next AbCT. Thus, the AAC incidence was 20%. Among the 17 who initially had AAC > 0, the AACS progression was 12/17 = 70.59% after being adjusted with the Hokanson method.

Analyses were performed and graphs were produced using SPSS software version 20 and GraphPad Prism version 9.3.1, respectively. A two-tailed p-value of 0.05 was considered to be statistically significant.

## 3. Results

### 3.1. Data Description of Participants in the Study

Table 4 below lists the patients’ epidemiological characteristics and data collected from the CCTA at the time of examination.

Table 5 below lists the data for continuous variables, presented as the mean with the standard deviations for normally distributed variables and the median with the interquartile range for non-normally distributed variables.

We summarized the demographic and clinical characteristics of our study in Table 4 and Table 5. Patients with psoriasis were of an upper-middle age (mean age ± SD 59 ± 13.5) and had moderate-to-severe skin disease (PASI score mean ± SD 10.57 ± 9.99 at the time of the CT scan). The two major psoriasis types were psoriasis vulgaris (53.0%) and psoriatic arthritis (38.6%). Most of the patients had suffered from psoriasis for a long time (duration mean ± SD 13 ± 10.42). The BMI of 62.6% of participants was classified as overweight or higher. In total, 56.6% of the patients in our study had codiagnosed hypertension, while only 16.9% were receiving diabetes treatment. More than half of the patients had dyslipidemia (53.0%). More than one-third of the patients had a history of smoking or alcohol consumption (37.3% and 43.4%, respectively). Other than hypertension, diabetes, and hyperlipidemia, 16.9% of the patients had a history of CVD, including 7.2% who suffered from CHD. Even though 73.4% of the patients were not satisfied with their diagnosis of metabolic syndrome, 14.5% of the participants had a history of family coronary disease, whereas a small percentage (2.4%) of individuals had a family history of psoriasis. Regarding the coronary morphology obtained from CCTA, the prevalence of noncalcified plaque was 12.0%, while calcified plaque was observed in 37.3% of the participants. Notably, 37.3% of the patients had been diagnosed with coronary stenosis, of which the three-quarters (74.2%) were cases of moderate-to-severe stenosis.

### 3.2. The Association between CCTA and CACS

In the current study, CACS was significantly higher in patients with one to two coronary plaques or more than in the group of patients with absolutely no coronary plaques (*p* < 0.001). Furthermore, the group of patients with five or more coronary plaques had a significantly higher CACS than the group of patients with only one to two plaques (*p* < 0.01) (Figure 3a). Similar to coronary plaques, the group of patients with one or more coronary stenosis had a significantly higher CACS than the group of patients without any stenosis at all (*p* < 0.001) (Figure 3b). When considering the degree of coronary stenosis, there was no significant difference in CACS between the group without stenosis and the group with mild stenosis nor was there a significant difference in CACS between the participants in the group with moderate-to-severe stenosis. However, there was a significant increase in CACS between the none-to-mild stenotic group and the moderate-to-severe stenosis group (Figure 3c). Notably, there was a strong correlation between CACS and the number of coronary plaques (Figure 3d), whereas no association was found between CACS and PASI score (Figure 3e). In terms of psoriasis subtypes, CACS was significantly higher in patients with psoriatic arthritis or pustular psoriasis than in those with psoriasis vulgaris (*p* < 0.01 and *p* = 0.03, respectively) (Figure 3f).

### 3.3. The Association between CCTA and AACS

A statistically significant increase in AACS was observed between the groups with three or more coronary plaques and those without (*p* < 0.05), but no statistically significant difference was observed in AACS between the participants in the groups with one to two plaques. There was also no statistically significant difference between the group with one to two plaques and the group with three or more plaques (Figure 4a). In terms of coronary stenosis number, there was only a significant increase in AACS in the group with three or more coronary stenoses compared to the group without stenosis (*p* < 0.01). No significant change in AACS was found in patients with one to two stenoses compared to those without any stenosis (Figure 4b). Interestingly, when considering the severity of coronary stenosis, the only significant difference in AACS was found between patients with moderate stenosis and those with no stenosis at all (*p* < 0.01) (Figure 4c). Similar to CACS, AACS was independent of the PASI score (Figure 4d). Furthermore, no differences in AACS were found between subtypes of psoriasis (Figure 4e).

### 3.4. The Correlation between CACS and AACS

The correlation coefficient between AACS and CACS was 0.6033 (*p* < 0.0001) (Spearman rank correlation). The linear regression equation between these two variables was Y = 0.04569 × U + 69.14, with AACS as the independent variable and CACS as the dependent measurement. There was a moderate correlation between AACS and CACS, which might reflect the alternative roles of AACS and CACS in certain critical outcomes.

### 3.5. Discrepancy of CACS and AACS among Subgroups of Variables

Regarding the differences between CACS and AACS across various subcategories, there was remarkable similarity between these two calcification scores from some viewpoints. In particular, the medians of both CACS and AACS were significantly higher in patients with plaques or any of the comorbidities than in those without plaques or comorbidities. Likewise, patients with hypertension also had significantly higher CACS and AACS than those with normal blood pressure. However, patients with diabetes had significantly higher AACS but not CACS compared to those without diabetes. Similarly, AACS demonstrated a significant difference among BMI classification subcategories, whereas CACS did not differ according to the patient’s body weight.

CACS might play a more major role than AACS in reflecting differences in other critical subgroups. CACS was significantly different in different psoriasis subtypes, while no similar difference was found in the case of AACS. CACS also gradually increased in groups with noncalcified plaques, mixed plaques, and calcified plaques, whereas no differences were noted with AACS. Although both CACS and AACS were statistically higher in patients with comorbidities, CACS but not AACS was found to be significantly different when we divided the patients into groups with and without CVD. It is noteworthy that CACS but not AACS was statistically lower in patients under anti-IL12/23 treatment than in patients not under anti-IL12/23 drug treatment. A history of familial coronary diseases was exploited as a variable in this study, and, surprisingly, neither CACS nor AACS was remarkably different in patients with a positive coronary family history when compared to that in patients without a family history.

CACS showed a consistently significant difference across subgroups of all the characteristics recorded with CCTA, such as the prevalence of coronary plaque, the coronary morphology, the prevalence of coronary calcified plaque, the prevalence of coronary stenosis, the degree of stenosis, and the number of involved coronary branches. Meanwhile, AACS was found to significantly differ only between the subgroups of coronary plaque, coronary calcified plaque, and coronary stenosis. This might imply that AACS is an indicator for the incidence of coronary plaque, especially calcified plaque, and coronary stenosis rather than a marker for further the involvement of coronary arteries as in the case of CACS.

### 3.6. The Role of CACS and AACS in Coronary Artery Involvement Diagnosis

Logistic regression was utilized to validate the role of CACS and AACS in the diagnosis of critical outcomes. The CACS cut-off point for calcified plaque diagnosis was chosen to be > 1.07, and the sensitivity and specificity were 100% and 94.2%, respectively (likelihood = 17.33 and J score = 94.2). The chosen CACS cut-off point for coronary stenosis prediction was >5.38, and the sensitivity and specificity were 80.6% and 86.5%, respectively (likelihood = 5.991 and J score = 67.2). CACS might be useful for comorbidities and CVD prediction. With its relatively high sensitivity and specificity, CACS was found to have high accuracy in calcified plaque and coronary stenosis predictions when the AUC for each outcome was 0.9888 (*p* < 0.0001) and 0.8567 (*p* < 0.0001), respectively. The CACS AUCs for comorbidities and CVD were moderate, at 0.7281 (*p* = 0.001) and 0.7019 (*p* = 0.018), respectively. AACS might also be utilized to predict the emergence of calcified plaque and coronary stenosis when the cut-off points of > 464.11 and > 43.65, respectively, are used. The relative sensitivity and specificity of AACS for coronary calcified plaque and coronary stenosis predictions were 72.0% and 79.5% for calcified plaque (likelihood = 3.51 and J score = 51.5), and 88.0% and 59.0% for coronary stenosis predictions (likelihood = 2.145 and J score = 47.0), respectively. These sensitivity and specificity were significantly lower than those for CACS. AACS possessed moderate AUC measurements in terms of calcified plaque and coronary stenosis plaque prediction, which were 0.8021 (*p* < 0.0001) and 0.7538 (*p* = 0.0007), respectively. AACS played a small role in comorbidity and CVD prediction.

### 3.7. The Progression of Artery Calcification and Its Correlation with Other Factors

While there were significant decreases in PASI score during each re-examination (Figure 5a), CACS did not significantly change during the follow-up time between the first and second CACS (Figure 5b). In contrast, AACS dramatically increased from the first time to the second time (Figure 5c; refer to Appendix A).

There was a significantly large difference in terms of CACS and AACS progression rates, which might imply progressive vascular calcification at the abdominal aorta in participants. Particularly, the AACS progression rate was much higher than the CACS progression rate in patients without hyperlipidemia. Furthermore, the AACS accumulation velocity in patients with comorbidities was also significantly higher than the CACS accumulation velocity in patients without any comorbidities.

Regarding the CACS and AACS progression rates and other factors, the higher the LDL cholesterol, the lower the CACS progression. Similarly, the AACS progression rate was in reversed linear regression with the plasma triglyceride levels of participants. AACS reduction occurred when the plasma triglyceride level was higher than 275 mg/dL.

## 4. Discussion

Concerning CACS, the results regarding our study’s patients with psoriasis are relatively consistent with those of other studies, for example, a study conducted on patients with psoriasis [20], a study on a healthy population in the Framingham Heart study [21], and study on American patients with asymptomatic CVD [22]. However, a remarkably higher prevalence and right-shift distribution of CACS have been noted in the studies conducted by Hisamatsu et al. [23] and Criqui et al. [24] in the MESA study population, which emphasized the role of participant recruitment criteria. Indeed, there was a consistency between the male group in our study and a study conducted by Ahuja et al. [25] as well as another study conducted by Hirata et al. [26] despite the younger healthy Japanese American male population in the study by Ahuja and the older healthy Japanese male population in the study by Hirata. This comparison suggests that CACS has a contributing effect on various ages, lifestyles, and levels of underlying psoriasis.

Regarding AACS, while our findings are consistent with those of Criqui et al. based on a healthy population in the MESA study [27], most other studies focusing on healthy populations have found a significantly lower prevalence and a left-shift distribution of AACS [9,21,28,29]. In terms of the association between CACS and AACS, the Spearman correlation coefficient for the pooled score of AAC and CAC was r_ho_ = 0.603, with *p* < 0.0001 (Figure 6 and Appendix A). This correlation was substantially stronger than those in multi-ethnic populations in previous studies (r = 0.38, *p* < 0.0001 [30], r = 0.39 [9]). Not only the correlation but also the distribution of both CACS and AACS in our study was substantially more spread out than those in the study conducted by Jurgens et al. [9]. Undoubtedly, the 95th percentiles of both CACS and AACS in our study were higher than those in the study conducted by Jurgens et al. (838.83–6089.17 compared to 200–1107, respectively) [9]. The 95th percentile of our study was also greater than the 90^th^ percentile of the calcification score in a study conducted by Criqui et al. [30]. All of this suggests that calcification progression in patients with psoriasis might be accelerated.

Our study showed that the PASI score, or the severity of psoriasis, is not associated with the calcification progression of either the coronary arteries or the abdominal aorta (Figure 5 and Appendix A). This association might reflect similar systemic exposing factors between these two parameters, although it could be postulated that there is no major association between the severity of psoriasis and the vascular calcification process. This result is also consistent with that of a previous paper, which showed no differences in CACS between patients with psoriasis and healthy volunteers [20]. However, the calcification process varied among different subtypes of psoriasis. CACS was significantly higher in patients with psoriatic arthritis and pustular psoriasis than in those with psoriasis vulgaris, whereas there was no significant difference in AACS. Therefore, there might be a selective coronary calcification in patients with psoriasis arthritis and pustular psoriasis regardless of PASI score, while no dominant calcification location was noted in psoriasis vulgaris.

In terms of coronary morphologies, we failed to demonstrate the role of either CACS or AACS in patients with noncalcified plaques, as confirmed by the CCTA result (Appendix A). Therefore, neither CACS nor AACS could be utilized as a tool for noncalcified plaque prediction. This result is in agreement with multiple publications that have made efforts to study individuals with high-risk noncalcified plaques [20,31,32]. On the other hand, both CACS and AACS could be used to predict the existence of coronary calcified plaque as alternative tests to CCTA because of their superior sensitivity and specificity.

CACS and AACS might also be applied as prediction tools for coronary stenosis. Both CACS and AACS were significantly higher among patients with coronary stenosis than among patients without coronary stenosis, as confirmed by the CCTA. We compared the roles of CACS and AACS in coronary stenosis prediction via AUC measurement. Our CACS result (AUC = 0.8567, *p* < 0.0001) (Figure 7b) was comparable to the AUC observed in previous papers (AUC = 0.81, *p* = 0.012 [33]; AUC = 0.84 [34]). Our AUC result of 0.8567 in patients with psoriasis was also similar to that in a publication by Kinoshita et al., who examined patients with and without diabetes (0.80 and 0.79, respectively) [35], although our cut-off point of CACS for coronary stenosis (5.38) was substantially lower than those of this author (50 and 40 for patients with and without diabetes, respectively). This distinction may be devoid of the variable definition. While we chose a mild, less than 50% lumen diameter reduction for CCTA coronary stenosis diagnosis regardless of noncalcified or calcified plaques, Kinoshita et al. discovered that a decrease of at least 50% in coronary artery lumen diameter was a significant stenosis outcome. CACS also differed between stenosis degrees and stenosis distribution across the main branches of coronary arteries. This suggested a potential role for CACS in coronary stenosis stratification. Our association between CACS and stenosis distribution is in agreement with the result of Rosen et al., who found the highest CACS in individuals with three-branch coronary stenosis [16]. Although our mean CACSs for each stenosis distribution subgroup were remarkably lower than those in the study conducted by Rosen et al. [16], it might be explained by our stricter stenosis definition.

Regarding the inconsistency between CACS and AACS in their association with the coronary stenosis confirmed by CCTA, AACS might only moderately reflect the status of coronary calcification, and it likely only partially reflects the calcification progress of the whole vascular bed. Several other factors might also contribute to the surge of AACS, such as systemic metabolism dysfunction, which might be specific to calcium deposition at the abdominal aorta. Indeed, in our study, we found that AACS, but not CACS, was significantly higher in patients with psoriasis with diabetes comorbidity (Table 6) in a similar manner to the way in which CACS, but not AACS, was specifically increased in psoriatic arthritis and pustular psoriasis (Figure 3f and Table 6). Furthermore, AACS was the highest in patients with psoriasis with other comorbidities but without CVDs (CHD, stroke, or arrhythmia) (Table 6). Similarly, AACS was not increased along with the increase in coronary plaque until the end, although it still rose relatively compared to its trajectory in those with no coronary plaque at all (Figure 4a). Therefore, there may no longer be a close association between AACS and the involvement noted by CCTA in the polar region of these two variables. The polar region of these two variables might be the place where some unclear factors are specifically working for the vascular calcification of each different location, and we would rather choose the more precise and direct parameters for patients in this polar region. Particularly in patients with various known damages to the CCTA, CACS accurately reflects the degree of involvement and is easier to follow during the follow-up examination than AACS. Otherwise, in patients with extremely high AACS results (more than 6000–8000), AACS at that time might no longer be an indicator for suspected widespread coronary damage. As a result, we prefer conducting a CACS or CCTA test at that time. In conclusion, although AACS represents an alternative means to predict coronary stenosis, it might not be suitable to distinguish between stenosis degrees and the spread of calcified coronary plaques.

We observed similarity between AACS and CACS in patients with and without CHD (Appendix A), which showed that neither CACS nor AACS played a role in CHD prediction. This result is opposite to the result of Jurgens et al., who demonstrated a specific role of CACS in a healthy population [9]. Therefore, a further investigation into which variables will be useful for CHD prediction in patients with psoriasis might be necessary. Our study, on the other hand, found a weak correlation between CACS—but not AACS—and CVDs, such as stroke or arrhythmia (Table 6 and Appendix A). The correlation between CACS and AACS and CVD in our study is aligned with that observed in a meta-analysis in individuals with suspected CVD [36], which showed the strongest association between the adjusted hazard ratios of major adverse cardiac events (MACE) and CACS > 400. Therefore, patients with psoriasis with greater CACS might also have an increased risk for MACE; a prospective cohort study might be necessary to confirm this hypothesis. However, our observation is opposite to that in the study conducted by Jurgens et al., which showed that AACS may play an alternative role in CVD predictions [9].

In terms of the body mass index (BMI) categories, AACS was highest in overweight participants—especially those younger than 55 years old—regardless of smoking status, while there was no noticeable difference in CACS across the subgroups (Table 6, Appendix A). We found this result to be in partial agreement with that of the study conducted by Rahman et al., who found that the BMI score associated with the highest point of AACS probability was around 25–26 and significantly reduced afterward, regardless of smoking status [37]. Despite that, a higher AACS in younger overweight psoriasis individuals has not been recorded before. Thus, this result allowed us to suspect the robust progress of atherosclerosis to become selectively calcified in the abdominal aorta in young overweight patients with psoriasis, which may be a result of the comprehensive effects of age and various cytokine complexes of both the chronic inflammatory state of psoriasis and overweight circumstances. Surprisingly, no similar phenomenon was recorded in coronary arteries. Furthermore, during the follow-up period, different calcification progressions were found between the coronary arteries and the abdominal aorta in terms of dyslipidemia and comorbidity (Figure 8 and Figure 9 and Appendix A). Therefore, a hypothesis might be proposed about slightly different vascular calcification pathogenesis in specific locations throughout the body in patients with psoriasis depending on the combined effects of age, BMI, dyslipidemia, and comorbidities. This dependence also reflects the complex effect of the interplay between psoriasis and the intrinsic factors of patients with psoriasis on the development of calcification plaques. Tobacco exposure may have a similar and simpler effect, as it has been shown that smokers are more likely to have selective peripheral artery disease than nonsmokers [38,39].

There have been studies that have clarified the link between CACS and hypertension [40,41]. Our results (Table 6 and Appendix A) are in accordance with those of several previous papers, which showed that the higher the systolic blood pressure, the more incidence and/or extension of CACS [19,30,40,42,43]. However, the impact of hypertension on CACS for patients with psoriasis became blurred when combined with their age (Appendix A). This could be due to the high prevalence and lifetime extension of CACS, AACS, and hypertension. Hence, both hypertension and the aging process might have a cumulative effect on vascular calcification.

AACS was significantly higher in hypertensive participants than in nonhypertensive participants in our study (Table 6), which is consistent with the results observed for hemodialysis patients [10] and men with abdominal aortic aneurysms [44] and for measurements taken using another method (aortic calcific deposit volume) [45]. For that reason, the increase in AACS in patients with psoriasis and hypertension might be similar to that in other special populations. In terms of AACS progression, a linear effect of neither systolic blood pressure nor diastolic blood pressure was found (data not shown), which is consistent with the result reported by Arai et al. [46]. Although O‘Donnell et al. found a weak positive correlation between systolic blood pressure and AACS (r = 0.27, *p* < 0.0001), AACS in this study was measured with lumbar radiographs, which might be inconsistent with our measurement method [47].

Patients with psoriasis in our study who also suffered from diabetes had significantly higher AACS than patients without diabetes (Table 6), which is consistent with previous publications [30,43,48]. However, the CACS result showed no significant difference between the subgroups of patients with diabetes. This result is contradictory to the previous literature [19,43,48], although it is somewhat consistent with the study conducted by Kanaya et al. [42], in which it was speculated that diabetes may play a role in a univariate model for CACS change, but the authors also determined it to no longer be a significant factor in a multivariable model. The CACS results obtained in the current study are also in accordance with those of the study conducted by Mansouri et al. [49], who showed that the psoriasis population did not have a remarkably higher CACS than individuals with diabetes. Moreover, we found a weak positive correlation between AACS and HbA1c (%) (r_ho_ = 0.271, *p* = 0.030) (Appendix A), which is slightly higher than that of European Americans (r = 0.13, *p* < 0.001) [50]. Although a similar association between CACS and HbA1c (%) could not be found, as in the case of Wagenknecht et al. and Kimani et al. [43,50], we did find a weak positive correlation between CACS and fasting glucose level (r_ho_ = 0.309, *p* = 0.006) (Appendix A), which is consistent with the higher prevalence of CACS together with an increase in glycemia [30]. Therefore, people with diabetes could probably have higher AACS and even CACS, but there might be more important factors contributing to the increase in CACS and AACS other than diabetes itself, especially in patients with psoriasis. Indeed, when we reclassified patients into age versus diabetes groups, the impact of diabetes became vague (Appendix A) or even no longer existed (Appendix A). In conclusion, diabetes might have an accumulative effect on psoriasis and might have a minor effect compared to the age of participants in vascular calcification.

The regression models for CACS change in participants with CACS greater than zero at the start of our study revealed that the standardized β-coefficients of psoriasis type was −0.772 (*p* = 0.003, 95%CI −0.528; −0.140). Therefore, psoriasis type, or psoriatic arthritis, is a particularly critical factor contributing to the progression of coronary calcification, which is in agreement with the high plaque burden in patients with psoriatic arthritis reported in previous papers [51,52,53]. Furthermore, blood lipid parameters contributed to both CACS and AACS acceleration rate in our study followed an inverse linear regression model (Figure 9). To the best of our knowledge, this current study adds to our understanding of AACS progression and its associated factors in patients with psoriasis.

## 5. Conclusions

Our study contributes to clarifying the degree of association between CACS and AACS in patients with psoriasis. This relationship could be due to the similar systemic exposure factors of these two parameters in individuals. It could be postulated from our results that there is no major association between the severity of psoriasis and vascular calcification progression. Nevertheless, psoriasis types—together with underlying conditions—play critical roles in the calcification process, as well as further cardiovascular disease burdens, in these patients. Thus, appropriate approach and multimethod screening strategies during follow-up examinations might be helpful for the early detection of atherosclerosis plaques in patients with psoriasis.

## Figures and Tables

**Figure 1 diagnostics-13-00274-f001:**
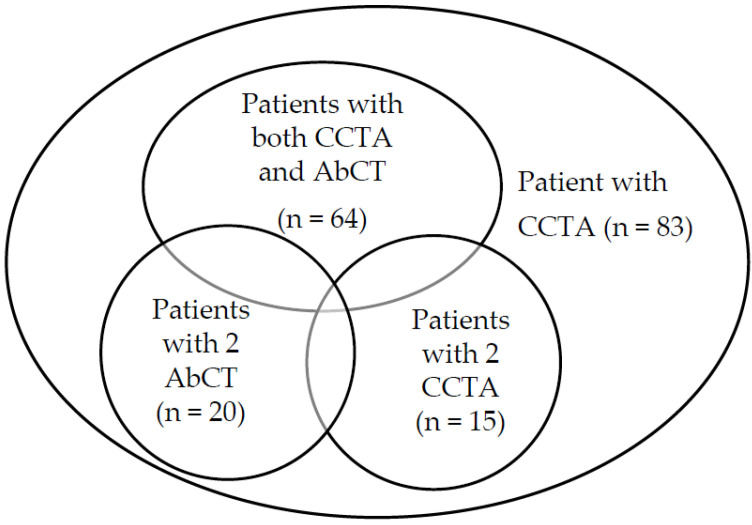
The sample population and the corresponding CACS, AACS results.

**Figure 2 diagnostics-13-00274-f002:**
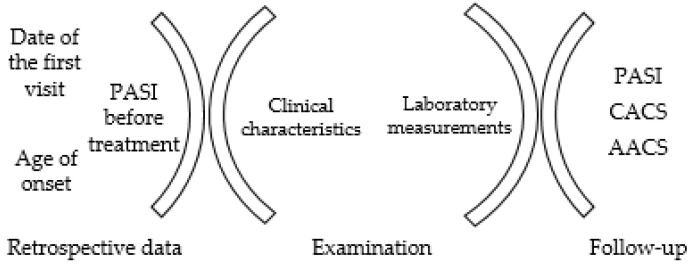
The progress of research sample collection.

**Figure 3 diagnostics-13-00274-f003:**
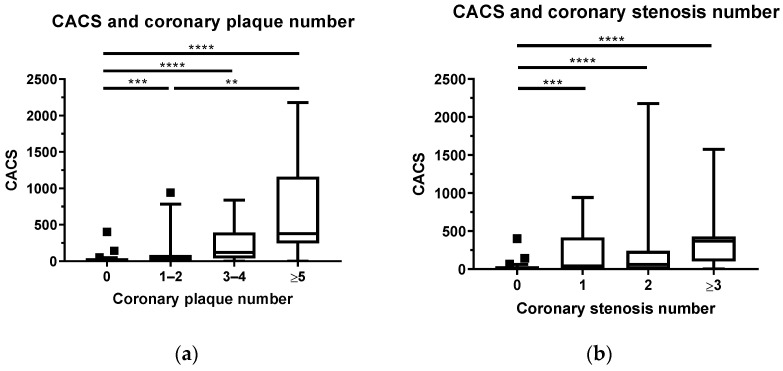
(**a**–**f**): The association between CCTA and CACS. (with *: *p* < 0.05, **: *p* < 0.01, ***: *p* < 0.001, ****: *p* < 0.0001) (■ CACS values exceeded 95th percentile of those subgroups; ● CACS value and the respective coronary plaque number; ○ CACS values and the respective PASI score).

**Figure 4 diagnostics-13-00274-f004:**
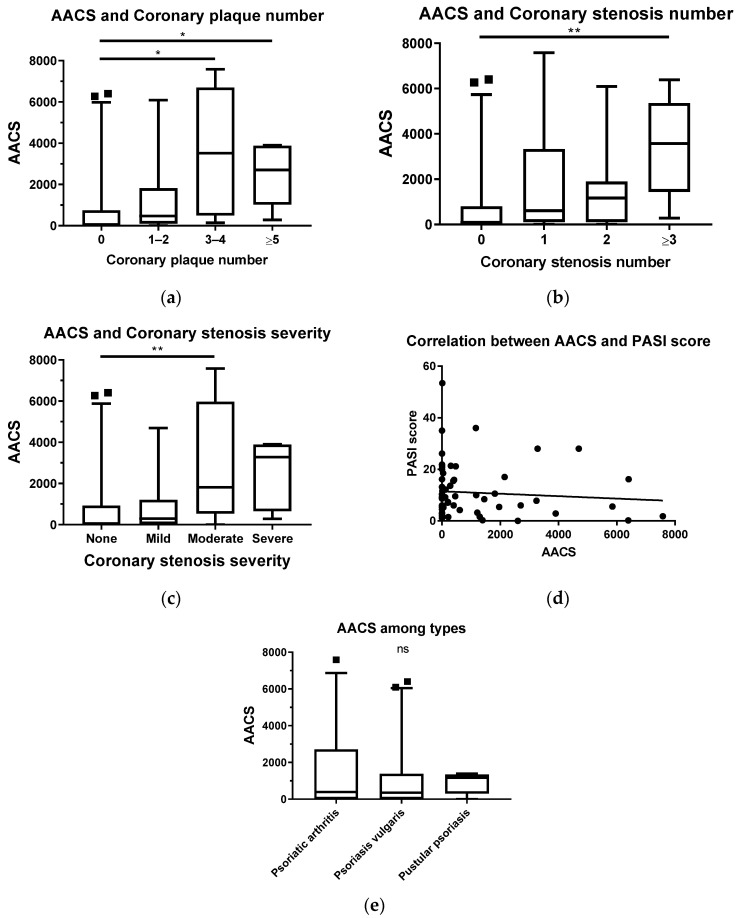
(**a**–**e**): The association between CCTA and AACS. (with *: *p* < 0.05, **: *p* < 0.01, “ns”: not significant) (■ AACS values exceeded 95th percentile of those subgroups; ● AACS values and the respective PASI scores).

**Figure 5 diagnostics-13-00274-f005:**
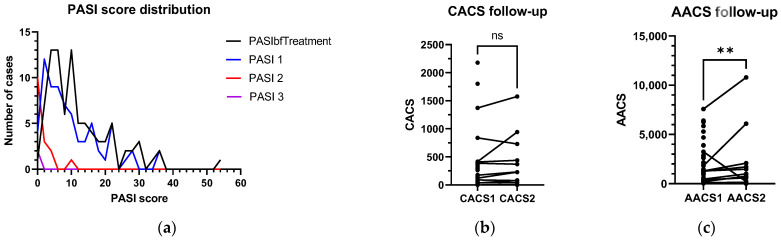
(**a**–**c**)**:** Progress of PASI score, CACS, and AACS during the follow-up period. (with **: *p* < 0.01, “ns”: not significant).

**Figure 6 diagnostics-13-00274-f006:**
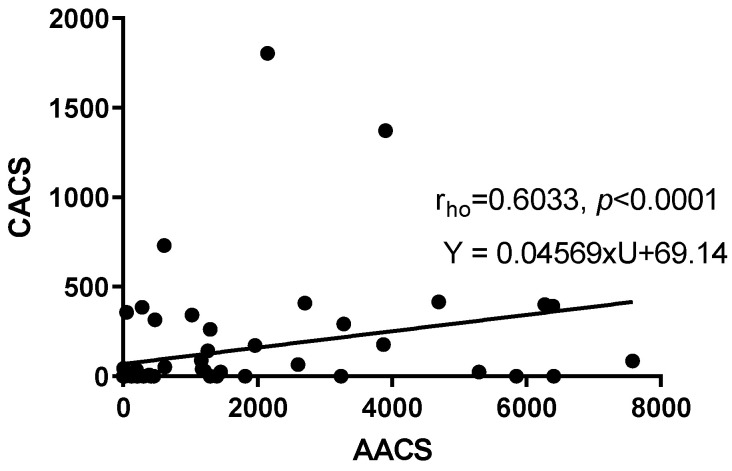
Spearman correlation between CACS and AACS.

**Figure 7 diagnostics-13-00274-f007:**
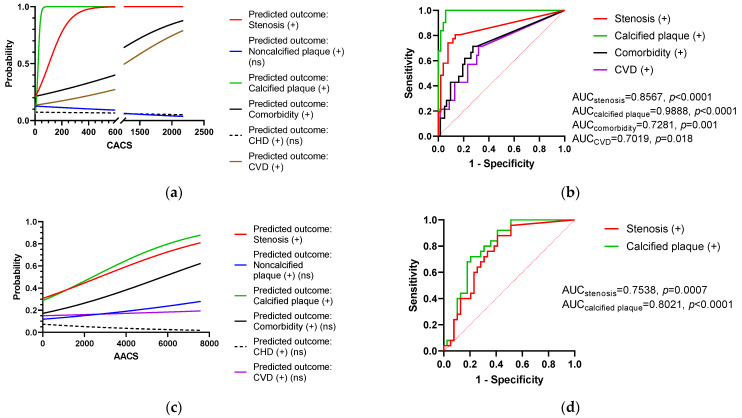
(**a**–**d**)**:** Value of CACS and AACS in the diagnosis of important outcomes and their relative sensitivities and specificities. (Red dotted line in (**b**–**d**): the AUC threshold is 0.5).

**Figure 8 diagnostics-13-00274-f008:**
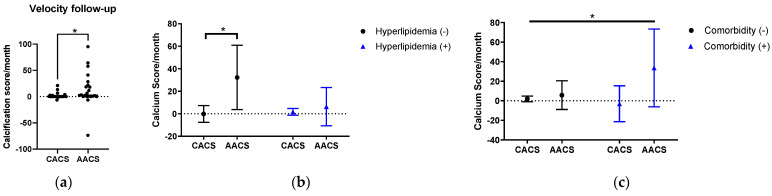
(**a**–**c**): Absolute calcification score progression during follow-up (pooled two-way ANOVA with *: *p* < 0.05).

**Figure 9 diagnostics-13-00274-f009:**
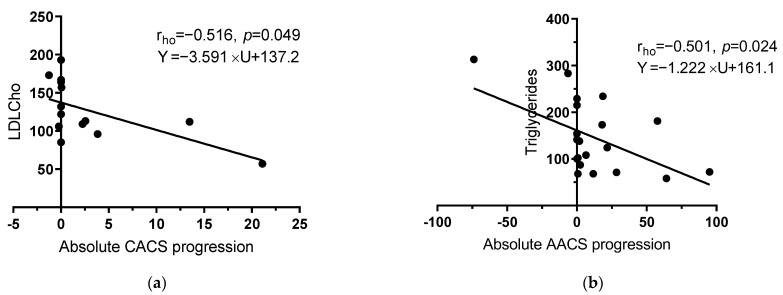
(**a**,**b**): Linear regression of absolute vascular calcification score progression and lipidemia parameters.

**Table 1 diagnostics-13-00274-t001:** Diagnostic criteria for metabolic syndrome.

Diagnostic Criteria		Cut Points
Elevated waist circumference		≥90 cm (men)
		≥80 cm (women)
Elevated TG		≥150 mg/dL
		or drug treatment for elevated TG
Reduced HDL-C		<40 mg/dL (men)
		<50 mg/dL (women)
		or drug treatment for reduced HDL-C
Elevated BP	Systolic BP	≥130 mmHg
	Diastolic BP	≥85 mmHg
		or current anti-hypertension treatment
Elevated fasting glucose		≥100 mg/dL
		or current anti-diabetic treatment

**Table 2 diagnostics-13-00274-t002:** Protocols of CT scanner parameters for CCTA and AbCT.

CT Scan Protocol	Tube Voltage (kVp)	Effective Tube Current (mAs)	Slice Thickness (mm)	Rotation Time (ms)	Threshold (Hounsfield Unit)	Field of View (Reconstruction) (mm)	Scan Collimation (mm)	Convolution Kernel	Reconstruction Matrix (mm)
Coronary	120	380	0.75	0.28	130	120–150	0.6	I36f ASA	512 × 512
Abdominal	120	350	5	0.5	130	330	1.0	Q40f	512 × 512

**Table 3 diagnostics-13-00274-t003:** Coronary stenosis criteria utilized in the study.

Coronary Stenosis Severity		Coronary Stenosis Distribution	
>0–49%	Mild	1-branch	Mild to severe stenosis of either the LAD, LCx, or RCA
50–74%	Moderate	2-branch	Mild to severe stenosis of two branches from the LAD, LCx, RCA
≥75%	Severe	Diffused (3-branch)	At the time of evaluation, a stenosis level was present in all three major branches.

**Table 4 diagnostics-13-00274-t004:** Demographic data at the time of examination.

Variables	Total	Category	Number (Percentage)
Gender	83	Male	59 (71.1)
		Female	24 (28.9)
Age	83	<55 years old	32 (38.5)
		55–64 years old	17 (20.5)
		65–74 years old	22 (26.5)
		≥75 years old	12 (14.5)
Psoriasis Type	83	Psoriasis vulgaris	44 (53.0)
		Psoriatic arthritis	32 (38.6)
		Pustular psoriasis	6 (7.2)
		Psoriasis erythroderma	1 (1.2)
BMI Classification	83	Underweight	2 (2.4)
		Normal	29 (34.9)
		Overweight	14 (16.9)
		Obesity class 1	29 (34.9)
		Obesity class 2	9 (10.9)
Comorbidity	83	Negative	62 (74.7)
		Coronary artery disease	6 (7.2)
		Stroke	4 (4.8)
		Arrhythmia	4 (4.8)
		Other	7 (8.5)
CHD	83	Positive	6 (7.2)
CVD	83	Positive	14 (16.9)
Smoking	83	Positive	31 (37.3)
Alcohol	83	Positive	36 (43.4)
Coronary Family History	83	Positive	12 (14.5)
Psoriasis Family History	83	Positive	2 (2.4)
Metabolic Syndrome	79	Negative	28 (35.4)
		Undefined	30 (38.0)
		Positive	21 (26.6)
Hyperlipidemia	83	Positive	44 (53.0)
Hypertension	83	Positive	47 (56.6)
Diabetes	83	Positive	14 (16.9)
CACS Classification	83	0	49 (59.0)
		1–10	4 (4.8)
		11–100	12 (14.5)
		101–400	11 (13.3)
		≥400	7 (8.4)
Coronary Morphology	83	Normal	44 (53.0)
		Noncalcified plaque	6 (7.2)
		Calcified plaque	25 (30.1)
		Mixed plaque	4 (4.9)
		Calcified plaque + coronary abnormality	2 (2.4)
		Coronary abnormalities only	2 (2.4)
Plaque	83	Positive	34 (41.0)
Noncalcified Plaque	83	Positive	10 (12.0)
Calcified Plaque	83	Positive	31 (37.3)
Stenosis	83	Positive	31 (37.3)
Suffering Vessel Distribution	83	None	49 (59.0)
		1 branch	13 (15.7)
		2 branches	13 (15.7)
		3 branches (diffused)	8 (9.6)
Stenosis Severity	83	None	52 (62.7)
		Mild	8 (9.6)
		Moderate	17 (20.5)
		Severe	6 (7.2)

**Table 5 diagnostics-13-00274-t005:** Clinical data and test results of participants.

**Variables**	**Number**	**Mean**	**Standard Deviation**	**95% Confidence Interval**		**Range**	
				**Lower**	**Upper**	**Minimum**	**Maximum**
Age	83	59	13.5	56	62	32	83
PASI before treatment	83	12.57	9.83	10.43	14.72	1.20	53.40
PASI before CT	70	10.57	9.99	8.19	12.95	0.00	53.40
Duration	79	13	10.42	10.59	15.26	0.3	41.0
Age of onset	80	46	16.97	42.3	49.85	14	78
BMI	83	25.0	4.21	24.08	25.92	17.58	43.28
Total cholesterol	82	188.67	64.02	174.60	202.74	0	300
Triglycerides	82	158.30	101.53	136.00	180.61	0	596
HDL cholesterol	73	53.48	13.96	50.22	56.74	27	98
LDL cholesterol	77	119.26	33.27	111.71	126.81	56	203
Systolic blood pressure	83	135	18.0	131.12	138.98	103	179
Diastolic blood pressure	83	82	14.5	78.98	85.31	42	119
**Variables**	**Number**	**Median**		**25–75 Percentile**		**Range**	
				**Lower**	**Upper**	**Minimum**	**Maximum**
CACS	83	0.00		0.00	66.02	0.00	2178.14
AACS	64	250.86		0.00	1423.21	0.00	7580.57
Plaque number	83	0		0	2	0	7
Stenosis number	83	0		0	1	0	4
Glucose	77	98.50		89.00	109.00	73.00	351.00
HbA1c	77	5.80		5.40	6.20	4.50	14.20
his-CRP	83	0.090		0.043	0.2778	0.014	7.216

**Table 6 diagnostics-13-00274-t006:** Differences between CACS and AACS across subcategories of variables.

Variables	Category	CACS				AACS			
		N	Mean	Median	*p*-Value	N	Mean	Median	*p*-Value
Age	<55 yo	32	5.08	0.00	<0.0001	25	156.84	0.00	<0.0001
	55–64 yo	17	148.90	24.03		16	1012.42	444.48	
	65–74 yo	22	86.29	0.00		15	1248.75	1023.71	
	≥75 yo	12	530.37	289.67		8	4803.17	4995.32	
Psoriasis Type	Psoriasis vulgaris	44	39.79	0.00	0.015	32	982.24	163.61	>0.05
	Psoriatic arthritis	32	249.89	32.39		27	1554.72	390.96	
	Pustular psoriasis	6	201.56	53.86		4	963.43	1229.98	
	Psoriasis erythroderma	1	0.00	0.00		1	13.54	13.54	
CACS Classification	0	49	0.00	0.00	<0.0001	37	602.98	0.00	<0.0001
	1–10	4	5.63	5.38		3	143.59	37.33	
	11–100	12	45.73	39.41		10	2130.53	1194.05	
	101–400	11	269.58	293.17		9	1500.26	1258.29	
	≥400	7	1059.99	838.83		5	3945.53	3903.99	
BMI Classification	Underweight	2	195.89	195.89	>0.05	1	0.00	0.00	0.003
	Normal	29	132.48	0.00		23	1079.40	37.33	
	Overweight	14	294.08	21.98		11	3377.40	2150.57	
	Obesity1	29	88.56	0.00		23	644.78	217.53	
	Obesity2	9	4.17	0.00		6	78.15	0.00	
Coronary Plaque	Negative	49	12.93	0.00	<0.0001	36	766.31	6.23	<0.0001
	Positive	34	303.62	87.23		28	1774.61	1194.05	
Coronary Morphology	Noncalcified plaque	6	0.00	0.00	0.001	6	1367.47	265.31	>0.05
	Mixed plaque	4	207.79	175.50		3	2333.15	1961.13	
	Calcified plaque	27	371.78	142.22		22	1911.51	1235.30	
Calcified Plaque	Negative	52	1.68	0.00	<0.0001	39	723.68	6.91	<0.0001
	Positive	31	350.62	173.08		25	1962.11	1258.29	
Stenosis	Negative	52	13.96	0.00	<0.0001	39	lesions	12.46	0.001
	Positive	31	330.03	122.16		25	1825.24	1175.81	
Stenosis Severity	Mild	8	84.31	4.03	0.021	8	695.35	444.48	>0.05
	Moderate	17	364.01	86.06		12	2474.04	1586.72	
	Severe	6	561.36	350.64		5	2075.93	2150.57	
Suffering Vessel Distribution	1 branch	13	64.36	24.03	0.009	12	968.18	403.52	>0.05
	2 branches	13	351.68	262.80		10	2452.79	1983.31	
	3 branches (diffused)	8	614.31	318.07		6	2257.18	2228.02	
Comorbidity	Negative	62	77.77	0.00	0.013	49	951.21	120.88	>0.05
	CHD	6	113.62	30.32		4	744.47	815.57	
	Stroke	4	265.10	110.79		3	1303.81	30.49	
	Arrhythmia	4	796.26	690.51		3	2148.51	2150.57	
	Other	7	172.52	86.06		5	3466.48	3280.24	
Comorbidity	Negative	62	77.77	0.00	<0.0001	49	951.21	120.88	0.028
	Positive	21	292.13	43.67		15	2044.49	1175.81	
CVD	Negative	69	87.38	0.00	0.008	54	1184.10	202.55	>0.05
	Positive	14	351.94	42.49		10	1333.49	815.57	
Hypertension	Negative	36	51.77	0.00	0.049	28	646.90	13.00	0.004
	Positive	47	193.46	0.00		36	1643.42	890.83	
Diabetes	Negative	69	137.14	0.00	>0.05	51	1077.69	114.53	0.032
	Positive	14	106.71	19.61		13	1716.50	1212.30	
Coronary Family History	Negative	71	151.78	0.00	0.050	53	1348.92	390.96	>0.05
	Positive	12	15.00	0.00		11	525.79	0.00	
Biologic Treatment	Without	62	169.01	0.00	0.013	51	1243.78	284.18	<0.0001
	With	16	6.05	0.00		9	241.08	5.80	
Anti-IL12/23 Treatment	Without	72	146.88	0.00	0.049	57	1147.11	206.33	>0.05
	With	6	0.00	0.00		3	72.51	0.00	

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
