# Peer review of "The Correlation between the Vascular Calcification Score of the Coronary Artery and the Abdominal Aorta in Patients with Psoriasis"

_diagnostics, 2023, doi:10.3390/diagnostics13020274_

Round 1

Reviewer 1 Report

Dear authors, congratulations on this interesting manuscript. I believe that some aspects will have to be clarified as follows:

- The authors mention that the study was approved by the Institutional Review Board of Kansai Medical University - please specify the number and date of the approval or a copy of the approval.

- Materials and Methods -must be positioned after the introduction, before the results. The results can be interpreted after studying the materials and methods.

- The discussions will have to be positioned before the conclusions.

- Results cannot start with tables. Please formulate an introductory phrase of the results, after which the tables can be inserted.

- I consider Table 1 to be difficult for the reader to follow. I recommend changing the three subsections (a, b, and c) either so that they all fit in one table or to make three distinct tables. Especially section c should be reorganized to be easier to follow.

- Also, in table 1, section c - the Categories use the initials O, A, P, and E  - I did not find the abbreviation explained in the text or at the end of the table.

- These categories are also in Table 2 - I did not find in the material and methods what represent these categories.

Author Response

[General Comment] Dear authors, congratulations on this interesting manuscript.

Response: Thank you very much for your high evaluation.

I believe that some aspects will have to be clarified as follows:

[Comment 1]: The authors mention that the study was approved by the Institutional Review Board of Kansai Medical University - please specify the number and date of the approval or a copy of the approval.

Response: Thank you for your kind reminder. We would like to add information about the number and date of the approval by the Institutional Review Board of Kansai Medical University for our study as follows:

“Informed consent was obtained from all subjects involved in the study (Kansai University of Medical Sciences Accreditation No. 2019195). The ethics review number for this research at Kansai Medical University is 2020148. The review was completed on November 25, 2020.”

[Comment 2]: Materials and Methods - must be positioned after the introduction, before the results. The results can be interpreted after studying the materials and methods.

Response: Thank you very much for your reminder. We feel very sorry for not following the template of the journal. We changed the order of the Material and Methods part before the Result part. The reference list was also changed accordingly.

[Comment 3]: The discussions will have to be positioned before the conclusions.

Response: Thank you very much for your reminder. The Discussion part was relocated after the Result part and before the Conclusions.

[Comment 4]: Results cannot start with tables. Please formulate an introductory phrase of the results, after which the tables can be inserted.

Response: Thank you very much for your feedback. I wrote the opening sentence before listing Table 1 to summarize the main content of Table 1. I have also added subsection titles so that you and the readers can follow the Results section more easily.

[Comment 5]: I consider Table 1 to be difficult for the reader to follow. I recommend changing the three subsections (a, b, and c) either so that they all fit in one table or to make three distinct tables. Especially section c should be reorganized to be easier to follow.

Response: Thank you very much for your valuable comment. We are very sorry for this inconvenience. We separated the content of table 1 into two distinct tables. The new table 1 lists all the category variables and their percentages for each category. The new table 2 lists all the continuous variables and their mean ± standard deviation for normally distributed variables (or their median with the interquartile range for non-normally distributed variables). The old Table 2 is named Table 3 after this adjustment.

[Comment 6]: Also, in table 1, section c - the Categories use the initials O, A, P, and E - I did not find the abbreviation explained in the text or at the end of the table.

Response: We highly appreciate your comment. We are very sorry for the miscommunication. We change the subcategories of the Type variable as follows: “Type O”, “Psoriasis vulgaris”; “Type A”, “Psoriatic arthritis”; “Type P”, “Pustular psoriasis”; “Type E”, “Psoriasis erythroderma”. Please note that Table 1 section c has been changed into Table 1 section a.

[Comment 7]: These categories are also in Table 2 - I did not find in the material and methods what represent these categories.

Response: Thank you very much for your minute observation. We changed the subcategories of the Type variable, as mentioned above. Please note that Table 2, which lists this variable data, has been changed into Table 3.

Reviewer 2 Report

Manuscript title:The correlation between vascular calcification score of coronary artery and abdominal aorta in psoriasis patients

Authors: Trang Nguyen-Mai Huynh et. al
 Recommendation

Would you be willing to review a revision of this manuscript? Comments to the Authors:

(A) Provide an overview/summary of the manuscript

This paper is a study that analyses the correlation between vascular calcification score of coronary artery and abdominal aorta in psoriasis patients by using the coronary artery calcification score(CACS) and abdominal aortic calcification score(AACS).

The topic of this article is of maximum interest because psoriasis patiens have a well known cardiovascular risk and a better assessment of it means a better outcome for the patients.

Thus, this article is very useful and of biggest interest for the specialists who take care of such patients.

The manuscript is structured in four parts: 1. Introduction

2. Results

3. Discussion

4. Material and Methods

A. Study design

B. Demographics, Clinical measurements, Laboratory measurements

C. Measurements of CACS and AACS by CT scan

D. Statistical analysis 5. Conclusions

 (1) Introduction

Trang Nguyen-Mai Huynh et. al highlighted the importance of corect assessment of cardiovascular risk in psoriasis patients. Patients suffering from psoriasis have a chronic systemic inflammation and mineral metabolism abnormalities which increase the risk of vascular calcifications. Coronary calcification assessment can be done by multiple studies, however, coronary CT angiography(CCTA) is chosen as a first line noninvasive modality, but, even this test is a difficult method. In this study, coronary artery calcification score (CACS) which is performed by a simple 3mm chest CT was verified in correlation to CCTA to evaluate the vascular calcification in these patients. The study also used the abdominal aortic calcification score (AACS) to make correlation between coronary artery calcification status and abdominal aortic calcification progression and the correlation between cardiovascular pathology and AACS.

(2) Results

CACS is significantly higher when having 1-2 coronary plaques or more compared to the group of patients with no coronary plaques. The group with 5 or more coronary plaques had significantly higher CACS than the group with only 1-2 plaques. The group with 1 or more coronary stenosis had a significantly higher CACS than the group of patients with no stenosis. There were no differences in CACS between the group with no stenosis and the group with mild stenosis and also no differences were seen between the group with moderate stenosis and severe stenosis. However there was a significant difference in the none to mild stenotic group and the moderate to severe stenosis group. There is a strong assosications between CACS and the number of plaques. There was no associations between CACS and PASI score. CACS was higher in patiens with psoriasis arthritis and pustular psoriasis compared to psoriasis vulgaris patients. It was only recorded a statistically significant increase in AACS between the group with 3 or more coronary plaques compared with the group without coronary plaques while there was no statistically significant difference in AACS between the

 group without coronary plaques and the group with 1-2 plaques. There was neither statistically significant difference between the group with 1-2 plaques and the group with 3 or more

plaques .In term of coronary stenosis numer, there was only a significant increase of AACS in the group with 3 or more coronary stenosis compare to the group without stenosis. No significant change of AACS was found in patients with 1-2 stenosis compared to those without any stenosis. Interestingly, when considering the severity of coronary stenosis, the only significant difference of AACS was found between patients with moderate stenosis and those with no stenosis at all .Similar to CACS, AACS was independent to PASI score. Not only that, no differences of AACS was found among subtypes of psoriasis.

In particular, the median of both CACS and AACS are much significantly higher in patients with plaques or any of comorbidities compared to those of patients without plaques or comorbidities. Likewise, patients with hypertension also had significantly higher CACS and AACS compared to those of patients with normal blood pressure. However, patients with diabetes had significant higher AACS but not CACS compared to those of patients without diabetes. Similarly, AACS showed a remarkable difference among subcategories of BMI classification whereas CACS was not different along patients’ body weight.

CACS but not AACS was statistically lower in patients under anti- IL12/23 treatment compared to those of patients without current anti-IL12/23 drugs.

Neither CACS nor AACS were remarkably different in patients with positive coronary family history compare to those of patients without family history.

(3) Discussion

The CACS and AACS results from this study are consistent with other studies conducted on psoriasis patients.

PASI score, or the severity of psoriasis, is not associated with the calcification progression of either coronary arteries or

 abdominal aorta, there is no major association between the severity of psoriasis and the vascular calcification progress,no differences of CACS between psoriasis and healthy volunteer. CACS was significantly higher in psoriasis arthritis and pustular psoriasis patients than those in psoriasis vulgaris whereas there was no significant difference of AACS.

In term of coronary morphologies, it was a failure to demonstrate the role of either CACS or AACS between patients with noncalcified plaques, to which the CCTA result can confirm . Therefore, neither CACS nor AACS could be utilized as tools for noncalcified plaque prediction. On the other hand, both CACS and AACS could be used to predict coronary calcified plaque existence as alternative tests for CCTA because of their superior sensitivity and specificity.

CACS and AACS might also be applied as prediction tools for coronary stenosis. Both CACS and AACS were significantly higher between patients with coronary stenosis compared to those of patients without coronary stenosis, confirmed by CCTA.

AACS might only moderately reflect the status of coronary calcification and probably only partially reflects the calcification progress of the whole vascular bed.

AACS, but not CACS, was found to be significantly higher in psoriasis patients with diabetes comorbidity .

In term of the body mass index (BMI) categories, AACS was highest in overweight participants, especially those younger than 55 years old regardless of smoking status, while there was no noticeable difference of CACS across the subgroups

AACS is significantly higher in hypertensive participants compared to those of nonhypertensive ones in our study

Psoriasis patients in our study who was co-suffered from diabetes had significantly higher AACS compared to those of the nondiabetes ones.

 Quality of the data presented is good and the results appear to be valid. The results reflect the method in organization and structure.

(4)Materials and methods

Materials and method are accurately described.

(5)Conclusion

Conclusions are supported by the data presented.

(6) Quality of English language

While reading the manuscript I have found multiple grammar errors which should be revised prior to acceptance.

One example of such mistake is ”1% of people all around the world are suffered from psoriasis ” .

Author Response

Manuscript title: The correlation between vascular calcification score of coronary artery and abdominal aorta in psoriasis patients

Authors: Trang Nguyen-Mai Huynh et. al
 Recommendation

Would you be willing to review a revision of this manuscript? Comments to the Authors:

(A) Provide an overview/summary of the manuscript

This paper is a study that analyses the correlation between vascular calcification score of coronary artery and abdominal aorta in psoriasis patients by using the coronary artery calcification score(CACS) and abdominal aortic calcification score(AACS).

The topic of this article is of maximum interest because psoriasis patiens have a well known cardiovascular risk and a better assessment of it means a better outcome for the patients.

Thus, this article is very useful and of biggest interest for the specialists who take care of such patients.

Response: Thank you very much for agreeing with us to the aim of this study.

The manuscript is structured in four parts:

  1. Introduction
  2. Results
  3. Discussion

  4. Material and Methods
  5. Study design
  6. Demographics, Clinical measurements, Laboratory measurements
  7. Measurements of CACS and AACS by CT scan
  8. Statistical analysis
  9. Conclusions

  (1) Introduction

Trang Nguyen-Mai Huynh et. al highlighted the importance of corect assessment of cardiovascular risk in psoriasis patients. Patients suffering from psoriasis have a chronic systemic inflammation and mineral metabolism abnormalities which increase the risk of vascular calcifications. Coronary calcification assessment can be done by multiple studies, however, coronary CT angiography(CCTA) is chosen as a first line noninvasive modality, but, even this test is a difficult method. In this study, coronary artery calcification score (CACS) which is performed by a simple 3mm chest CT was verified in correlation to CCTA to evaluate the vascular calcification in these patients. The study also used the abdominal aortic calcification score (AACS) to make correlation between coronary artery calcification status and abdominal aortic calcification progression and the correlation between cardiovascular pathology and AACS.

(2) Results

CACS is significantly higher when having 1-2 coronary plaques or more compared to the group of patients with no coronary plaques. The group with 5 or more coronary plaques had significantly higher CACS than the group with only 1-2 plaques. The group with 1 or more coronary stenosis had a significantly higher CACS than the group of patients with no stenosis. There were no differences in CACS between the group with no stenosis and the group with mild stenosis and also no differences were seen between the group with moderate stenosis and severe stenosis. However, there was a significant difference in the none to mild stenotic group and the moderate to severe stenosis group. There is a strong assosications between CACS and the number of plaques. There were no associations between CACS and PASI score. CACS was higher in patiens with psoriasis arthritis and pustular psoriasis compared to psoriasis vulgaris patients. It was only recorded a statistically significant increase in AACS between the group with 3 or more coronary plaques compared with the group without coronary plaques while there was no statistically significant difference in AACS between the group without coronary plaques and the group with 1-2 plaques. There was neither statistically significant difference between the group with 1-2 plaques and the group with 3 or more plaques. In term of coronary stenosis numer, there was only a significant increase of AACS in the group with 3 or more coronary stenosis compare to the group without stenosis. No significant change of AACS was found in patients with 1-2 stenosis compared to those without any stenosis. Interestingly, when considering the severity of coronary stenosis, the only significant difference of AACS was found between patients with moderate stenosis and those with no stenosis at all. Similar to CACS, AACS was independent to PASI score. Not only that, no differences of AACS was found among subtypes of psoriasis.

In particular, the median of both CACS and AACS are much significantly higher in patients with plaques or any of comorbidities compared to those of patients without plaques or comorbidities. Likewise, patients with hypertension also had significantly higher CACS and AACS compared to those of patients with normal blood pressure. However, patients with diabetes had significant higher AACS but not CACS compared to those of patients without diabetes. Similarly, AACS showed a remarkable difference among subcategories of BMI classification whereas CACS was not different along patients’ body weight.

CACS but not AACS was statistically lower in patients under anti- IL12/23 treatment compared to those of patients without current anti-IL12/23 drugs.

Neither CACS nor AACS were remarkably different in patients with positive coronary family history compare to those of patients without family history.

(3) Discussion

The CACS and AACS results from this study are consistent with other studies conducted on psoriasis patients.

PASI score, or the severity of psoriasis, is not associated with the calcification progression of either coronary arteries or abdominal aorta, there is no major association between the severity of psoriasis and the vascular calcification progress, no differences of CACS between psoriasis and healthy volunteer. CACS was significantly higher in psoriasis arthritis and pustular psoriasis patients than those in psoriasis vulgaris whereas there was no significant difference of AACS.

In terms of coronary morphologies, it was a failure to demonstrate the role of either CACS or AACS between patients with noncalcified plaques, to which the CCTA result can confirm. Therefore, neither CACS nor AACS could be utilized as tools for noncalcified plaque prediction. On the other hand, both CACS and AACS could be used to predict coronary calcified plaque existence as alternative tests for CCTA because of their superior sensitivity and specificity.

CACS and AACS might also be applied as prediction tools for coronary stenosis. Both CACS and AACS were significantly higher between patients with coronary stenosis compared to those of patients without coronary stenosis, confirmed by CCTA.

AACS might only moderately reflect the status of coronary calcification and probably only partially reflects the calcification progress of the whole vascular bed.

AACS, but not CACS, was found to be significantly higher in psoriasis patients with diabetes comorbidity.

In term of the body mass index (BMI) categories, AACS was highest in overweight participants, especially those younger than 55 years old regardless of smoking status, while there was no noticeable difference of CACS across the subgroups

AACS is significantly higher in hypertensive participants compared to those of nonhypertensive ones in our study

Psoriasis patients in our study who was co-suffered from diabetes had significantly higher AACS compared to those of the nondiabetes ones. 

 Quality of the data presented is good and the results appear to be valid. The results reflect the method in organization and structure.

Response: Thank you very much for your appreciation.

(4) Materials and methods

Materials and method are accurately described.

(5) Conclusion

Conclusions are supported by the data presented.

(6) Quality of English language

While reading the manuscript I have found multiple grammar errors which should be revised prior to acceptance.

One example of such mistake is ”1% of people all around the world are suffered from psoriasis”.

Response: Thank you very much for your review. Your review section is very detailed and we feel very grateful for your evaluation. Regarding the grammar and spelling errors, we are very sorry for the inconvenience. I have edited the entire manuscript with the support of a proofreading service. We hope the revised manuscript will meet your criteria.
